# The Combination of In Vitro Assessment of Stress Tolerance Ability, Autoaggregation, and Vitamin B-Producing Ability for New Probiotic Strain Introduction

**DOI:** 10.3390/microorganisms10020470

**Published:** 2022-02-19

**Authors:** Natalya Yu. Khromova, Julia M. Epishkina, Boris A. Karetkin, Natalia V. Khabibulina, Andrey V. Beloded, Irina V. Shakir, Victor I. Panfilov

**Affiliations:** Biotechnology Department, Mendeleev University of Chemical Technology, 125047 Moscow, Russia; khromova.n.i@muctr.ru (N.Y.K.); epishkina.y.m@muctr.ru (J.M.E.); khabibulina.n.v@muctr.ru (N.V.K.); beloded.a.v@muctr.ru (A.V.B.); ishakir@muctr.ru (I.V.S.); vip@muctr.ru (V.I.P.)

**Keywords:** probiotics, *Bifidobacterium*, *Lactobacillus*, probiotic properties, water-soluble B vitamins, biofortification, stress tolerance ability, aggregation ability

## Abstract

The health benefits of probiotics are beyond doubt. The positive effects of lactobacilli and bifidobacteria on the function of many body systems have been repeatedly proven by various studies. To completely realize the potential of probiotic microorganisms, the strains should be tested by the greatest combination of characteristics that contribute to the wellness of the host. In this work, for the first time, a combined assessment of the probiotic properties and vitamin B-producing potential of various species and strains of bifidobacteria and lactobacilli was carried out. The presence of an additional advantage, such as vitamin-producing ability, can prevent vitamin deficiency both at the level of the consumption of fermented foods, when the enrichment will occur naturally on the spot, and during colonization by these intestinal strains, when synthesis will occur in vivo. To select potential probiotics, the stress tolerance ability of 16 lactic acid bacteria and bifidobacteria strains to low pH values, bile, and proteolytic enzymes, as well as their ability to autoaggregate, were studied under conditions of modeling the gastrointestinal tract in vitro. The ability of the strains to extracellularly accumulate water-soluble B vitamins was evaluated by capillary electrophoresis. Among the tested strains of bifidobacteria, *B. adolescentis* VKPM AC-1662 is of interest; it was characterized by the greatest stress tolerance ability and the ability to autoaggregate, in addition to the extracellular synthesis of riboflavin and pyridoxine. Among lactic acid bacteria, *L. sakei* VKPM B-8936 demonstrated the greatest tolerance to low pH, *L. plantarum* VKPM B–11007 to duodenal conditions, *L. acidophilus* VKPM B-2213 to pepsin, and *L. salivarius* VKPM B–2214 to pancreatin. The highest percentage of autoaggregation was observed in *L. salivarius* VKPM B-2214, which also accumulated the largest amount of pantothenic acid, but it was sensitive to stress conditions. The obtained results could be used to create new products enriched with probiotics and B vitamins.

## 1. Introduction

Fortification of food products with chemically synthesized vitamins is a traditional way of increasing their biological value in order to prevent vitamin deficiency, which can have negative effects on the development and functioning of the body. A lack of B vitamins in humans can lead to various diseases. Folic acid deficiency can lead to osteoporosis, coronary heart disease, decreased cognitive function and Alzheimer’s disease, hearing loss [1], and an increased risk of breast cancer [2]. Riboflavin deficiency can lead to skin and liver damages, as well as changes in glucose metabolism in the brain [3]. Thiamine deficiency causes beriberi disease, which can have consequences for both the cardiovascular and the nervous system. Niacin deficiency can lead to pellagra, lack of pantothenic acid to acne, pyridoxine deficiency to dermatitis and epileptic seizures, and a lack of biotin can cause growth retardation in children [4]. Recently, the efficacy of synthetic vitamins has been in doubt and so they need to be analyzed. It is reported that their bioavailability is worse than that of natural forms of vitamins, which means they might not be as effective [5]. Also, some studies show that chemically synthesized vitamins can even have a negative effect on human health. For example, high consumption of foods enriched with chemically synthesized folic acid, not the natural form, in some people can lead to masking vitamin B12 deficiency, leukemia, arthritis, and an increased risk of cancer [6]. At the same time, synthetic vitamins are quite expensive due to the large number of stages of production. Finally, consumer requirements for food quality are increasing and preference is given to a healthy diet, so much attention is paid to the absence of synthetic additives in the composition.

Biofortification by fermentation of lactic acid bacteria and bifidobacteria with vitamin-producing ability is a promising approach for obtaining foods with increased concentrations of natural forms of B vitamins. Most members of the *Lactobacillus* and *Bifidobacterium* are normal representatives of the human intestinal microbiota and are generally recognized as safe (GRAS), which makes them the most suitable candidates for these purposes. Some authors insist that careful selection of starter cultures with increased vitamin B-synthesizing ability can enhance the concentration of folic acid in yogurts over 200 μg L^−1^ [7,8,9] or beverages, and thereby lead to the development of innovative functional products [10]. Such foods will not only be healthy, but also economically beneficial. The strains of lactic acid bacteria (LAB) selected according to the specified criteria have already been used as starters for bread and pasta production enriched with riboflavin, which leads to an increase in this vitamin concentration of about 2–3-fold in the final product [11]. It was shown in model rats with folic acid deficiency that yogurt fermented with the vitamin-producing strain *L. plantarum* GSLP-7 V restores the disturbed microbiota and increases levels of serum folate and homocysteine, while folic acid aggravated intestinal dysbiosis [12]. Moreover, vitamin production is considered a criterion in the functionality assessment in some studies—for example, on isolation of new LAB from various sources such as vegetable raw materials [13].

Vitamin-producing lactic acid bacteria and bifidobacteria strains may become potential candidates for their synthesis in vivo in the intestine after colonization [14]. It is reported that LAB synthesizing folate and riboflavin can be used as an additional treatment in people suffering from inflammatory bowel diseases [15]. To ensure the effects, LAB have to reach the colon in a viable state and be characterized by high stress tolerance when passing through the digestive tract (tolerance to acidic pH, bile, and proteolytic enzymes). These abilities refer to probiotic properties [16]. In addition to stress tolerance, the strains should have a high aggregation ability, which is often associated with adhesion to the intestinal epithelium and colonization properties [17]. They will have more advantages over nonaggregated strains that can be easily removed from the intestinal environment [18].

In this study, for the first time, the probiotic properties and B-vitamin-producing potential of various bifidobacteria and lactobacilli strains are evaluated in order to select the most promising candidates. Their stability under conditions that model the human gastrointestinal tract passage, autoaggregation ability, and ability to synthesize and extracellularly accumulate water-soluble B vitamins are studied. The obtained results can be used in medicine, healthcare, the food industry, and farming in order to expand the applications of LAB and bifidobacteria.

## 2. Materials and Methods

### 2.1. Bacterial Strains and Growth Conditions

The bifidobacteria strains *B. adolescentis* AC-1662, *B. pseudolongum* subsp. *pseudolongum* AC-1785, *B. longum* subsp. *infantis* AC-1912, *B. breve* AC-1911, *B. longum* subsp. *longum* AC-1665, and *B. bifidum* AC-1779 and lactic acid bacteria *L. acidophilus* B-2105, *L. plantarum* B-11007, *L. acidophilus* B-6551, *L. acidophilus* B-2213, *L. salivarius* B-2214, *L. casei* B-2873, *L. acidophilus* B-6553, *L. rhamnosus* B-8238, *L. sakei* B-8936, and *L. paracasei* subsp. *paracasei* B-4079 were used in this study. Freeze-dried samples were purchased from the Russian National Collection of Industrial Microorganisms (VKPM, Moscow, Russia) in sealed glass vials. The sources of origin of microorganisms, according to VKPM, are presented in Table 1.

To obtain the inocula, bifidobacteria were inoculated in a Bifidum medium [19] and incubated at 37 °C for 24 h. The bifidum medium has the following composition (Formula Per Liter): 30 g casein tryptone (211610; Difco Laboratories, Detroit, MI, USA); 5.0 g yeast extract (Springer, Maisons-Alfort, France); 10.0 g D-glucose (361105; Dextrose, Roqquette, Lestrem, France); 0.5 g cysteine hydrochloride (C7477; Sigma-Aldrich, Munich, Germany); 2.5 g sodium chloride (S9888; Sigma-Aldrich, Munich, Germany); 0.5 g magnesium sulfate (212486; PanReac Applichem, Barcelona, Spain); 0.5 g ascorbic acid (A4034; Dia-m, Moscow, Russia); and 0.3 g sodium acetic acid (S2889; Sigma-Aldrich, Munich, Germany).

To obtain the inoculate of lactobacilli, the strains were cultured in a de Man–Rogosa–Sharpe (MRS) broth [20]. To determine the ability of bifidobacteria and lactobacilli to produce B vitamins, a folic acid assay medium was used according to the Difco™ formula with some modifications in the composition (approximate formula per liter): 12.0 g casamino acids (tryptone) (T2559; Sigma-Aldrich, Munich, Germany); 40.0 g dextrose (361105; Roqquette, Lestrem, France); 20.0 g sodium citrate (141655; PanReac Applichem, Barcelona, Spain); 0.2 g L-cystine (C8755; Sigma-Aldrich, Munich, Germany); 0.2 g DL-tryptophan (T3300; Sigma-Aldrich, Munich, Germany); 20.0 mg adenine sulfate (145815; Sigma-Aldrich, Munich, Germany); 20.0 mg guanine (G11950; Sigma-Aldrich, Munich, Germany); 20.0 mg uridine (U3003; Sigma-Aldrich, Munich, Germany); 2.0 mg thiamine hydrochloride (W332208; Sigma-Aldrich, Munich, Germany); 4.0 mg pyridoxine hydrochloride (P6280; Sigma-Aldrich, Munich, Germany); 2.0 mg riboflavin (R9504; Sigma-Aldrich, Munich, Germany); 2.0 mg niacin (N4126; Sigma-Aldrich, Munich, Germany); 200.0 μg *p*-aminobenzoic acid (A9580; Sigma-Aldrich, Munich, Germany); 0.8 μg biotin (19606; Sigma-Aldrich, Munich, Germany); 1.0 g dipotassium phosphate (141512; PanReac Applichem, Barcelona, Spain); 1.0 g monopotassium phosphate (141509; PanReac Applichem, Barcelona, Spain); 0.4 g magnesium sulfate (142486; PanReac Applichem, Barcelona, Spain); 20.0 mg sodium chloride (S9888; Sigma-Aldrich, Munich, Germany); 20.0 mg ferrous sulfate (1413624 PanReac Applichem, Barcelona, Spain); and 20.0 mg manganese sulfate (131413; PanReac Applichem, Barcelona, Spain) (pH adjusted to 7.0).

### 2.2. Tolerance to Simulated Human GI Tract

The cultures were centrifuged (Eppendorf, Hamburg, Germany Centrifuge 5417C, 7000 rpm, 15 min) after 24 h of growth (daily culture) and washed once in a 1:1 *v*/*v* ratio with a Ringer solution of the following composition (g L^−1^): 8.69 sodium chloride (S9888; Sigma-Aldrich, Munich, Germany); 0.30 potassium chloride (191494; PanReac Applichem, Barcelona, Spain); 0.48 calcium chloride (131232; PanReac Applichem, Barcelona, Spain), pH 6.4 [21]. The precipitate was resuspended in simulated gastric juices (see below) in a 1:1.33 ratio [22] and incubated at 37 °C. Sampling was carried out for 20, 40, 60, and 90 min of the experiment for bifidobacteria and for 30, 60, and 90 min for lactobacilli and the viable cells was counted. After 90 min, the samples were centrifuged, the supernatant was decanted, and a medium simulating the conditions of the duodenum (see below) was added in a 1:3 *v*/*v* ratio [22]. If the strain was found to completely lose viability in gastric juices, the simulated intestinal conditions test was repeated with fresh daily culture. Samples were taken at 60, 120, and 180 min.

The effect of the bile medium on the viability of bifidobacteria and lactobacilli was also studied [22]. The precipitates were obtained as described above and resuspended in a bile medium (1:1). The incubation was carried out at 37 °C with sampling at 60, 120, and 180 min.

#### 2.2.1. Preparation of Simulated Gastric Juices

During the experiment, simulated gastric juices of the following composition (g L^−1^) were used: 3.50 dextrose (361105; Roqquette, Lestrem, France); 2.05 sodium chloride (S9888; Sigma-Aldrich, Munich, Germany); 0.60 potassium dihydrogen phosphate (A1043; PanReac Applichem, Barcelona, Spain); 0.11 calcium chloride (131232; PanReac Applichem, Barcelona, Spain); and 0.37 potassium chloride (191494; PanReac Applichem, Barcelona, Spain), pH 2.0. The solution was autoclaved at 115 °C for 20 min. The following components were sterilized by filtration (0.2 μm, Millipore, St. Louis, MO, USA) and added to the solution after cooling: 0.05 g L^−1^ pork bile (48305; Sigma, St. Louis, MO, USA); 0.10 g L^−1^ lysozyme (CAS Number 9066-59-5; Caglificio Clerici S.p.A., Cadorago, Italy), and 13.30 mg L^−1^ pepsin (12762; JSC “Plant of Endocrine Enzymes”, Moscow, Russia) [23].

#### 2.2.2. Preparation of Simulated Duodenal Environment

During the experiment, a medium [24,25] of the following composition (g L^−1^) was used: 6.43 sodium chloride (S9888; Sigma-Aldrich, Munich, Germany); 0.50 potassium chloride (191494; PanReac Applichem, Barcelona, Spain); 0.21 calcium chloride (131232; PanReac Applichem, Barcelona, Spain); 0.05 potassium dihydrogen phosphate (A1043; PanReac Applichem, Barcelona, Spain); 0.05 magnesium chloride (131396; PanReac Applichem, Barcelona, Spain); 5.67 sodium hydrogen carbonate (131638; PanReac Applichem, Barcelona, Spain); 0.15 urea (U5378; Sigma-Aldrich, Munich, Germany); albumin (CAS Number 9048-46-8; “Sonac Loenen BV”, GD Loenen, Netherlands); 10.00 pork bile (48305; Sigma, St. Louis, MO, USA); 1.00 lipase (CAS Number 9001-62-1; Caglificio Clerici S.p.A., Cadorago, Italy); and 6.00 pancreatin (A0585; Panreac, Barcelona Spain), pH 6.0.

#### 2.2.3. Preparation of Bile Medium

During the experiment, a medium [26] of the following composition was used: 30 mL of sodium chloride (S9888; Sigma-Aldrich, Munich, Germany), 175.3 g L^−1^ solution; 68.3 mL of sodium hydrogen carbonate (131638; PanReac Applichem, Barcelona, Spain), 84.7 g L^−1^ solution; 4.2 mL of potassium chloride (191494; PanReac Applichem, Barcelona, Spain), 89.6 g L^−1^ solution; 200 mL of hydrochloric acid (170266; LenReaktiv, Moscow, Russia), 37% w/v solution; 10 mL of urea (U5378; Sigma-Aldrich, Munich, Germany), 25 g L^−1^ solution; 10 mL of calcium chloride dihydrate (121214; PanReac Applichem, Barcelona, Spain), 22.2 g L^−1^ solution; albumin (CAS Number 9048-46-8; “Sonac Loenen BV,” Netherlands), 1.8 g L^−1^; and pork bile (48305; Sigma, St. Louis, MO, USA), 6.0 g L^−1^, pH 8.0.

#### 2.2.4. Tolerance to Enzyme Preparations

The tolerance of microorganisms to pepsin and pancreatin was evaluated separately. The bacterial cell precipitate was resuspended in the buffered enzyme (pancreatine or pepsin) solution in a 1:1.33 ratio. The enzyme solutions were previously sterilized by filtration (0.22 μm, Pall Corporation, Port Washington, NY, USA) [22,27].

To obtain the pancreatin (A0585; Panreac Applichem, Barcelona, Spain) solution (6 g L^−1^), a phosphate buffer of the following composition (g L^−1^) was used: 8.00 sodium chloride (S9888; Sigma-Aldrich, Munich, Germany); 0.24 sodium hydrogen phosphate (131679; PanReac Applichem, Barcelona, Spain); 0.20 potassium chloride (191494; PanReac Applichem, Barcelona, Spain); 0.24 potassium dihydrogen phosphate (A1043; PanReac Applichem, Barcelona, Spain) [28], pH 7.0. Pepsin (12762; JSC “Plant of Endocrine enzymes”, Moscow, Russia) was dissolved in the citrate–HCl buffer (pH 3.0) at 13.5 g L^−1^ [23,25]. To prepare the citrate–HCl buffer at pH 3.0, 21.0 g of citric acid (C2404; Sigma-Aldrich, Munich, Germany) was dissolved in 200 mL of 1 M NaOH, brought up to 1 L with distilled water, and 40.3 mL of this solution was diluted with 0.1 M HCl to 100.0 mL [29].

### 2.3. Determination of Total Viable Count and Survival Percentage

Enumeration of *Lactobacillus* strains viable cells (CFU mL^−1^) was carried out by plating 10-fold dilutions of the samples in sterile PBS on MRS agar. Plates were incubated at 37 ± 1 °C for 48 h. Enumeration of *Bifidobacterium* strains’ viable cells (CFU mL^−1^) was carried out by plating 10-fold dilutions of the samples in sterile PBS on Bifidum agar. Plates were incubated under anaerobic conditions provided by the BD GasPak™ (BD Biosciences, Franklin Lakes, NJ, USA) anaerobic container system at 37 ± 1 °C for 48 h. Counting was performed in triplicate and the results are expressed as log_10_ (CFU mL^−1^).

The survival percentage was calculated as the percentage of LAB or *Bifidobacterium* colonies grown on MRS or Bifidum agar compared to the initial bacterial concentration:(1)$$\%\;\mathrm{survival}\;\mathrm{percentage}=\frac{\log\;\mathrm{CFU}\;\mathrm{of}\;\mathrm{viable}\;\mathrm{cells}\;\mathrm{survived}}{\log\;\mathrm{CFU}\;\mathrm{of}\;\mathrm{initial}\;\mathrm{viable}\;\mathrm{cells}\;\mathrm{inoculated}}\times 100.$$

### 2.4. Spectrophotometry Autoaggregation Assays

The ability to autoaggregate was tested as described by Collado, Meriluoto, and Salminen [18] according to the autoaggregate percentage. The cells were harvested after 24 h of incubation by centrifugation (3200× *g*, 4 °C, 20 min), washed twice with a phosphate–salt buffer (PBS: 130 mM sodium chloride, 10 mM sodium phosphate, pH 7.2), resuspended in the same buffer under sterile conditions, adjusted to OD_600_ = 0.25 ± 0.05, and incubated at 37 °C. OD was controlled at different time intervals (0, 3, 6, 12, or 24 h). The percentage of autoaggregation was expressed according to:(2)$$A\%=\left(A_{0}-A_{t}\right)/A_{0}*100,$$
where $A_{0}$ is the absorption ($A_{600\;nm}$) at 0 h and $A_{t}$ is the absorption ($A_{600\;nm}$) at different time intervals (0, 3, 6, 12, or 24 h).

### 2.5. Determination of Water-Soluble B Vitamins

The analysis of water-soluble B vitamins free forms and their concentrations in the fermentation broth of probiotic bacteria was carried out with a capillary electrophoresis system “Kapel-105M” (Lumex Ltd., St. Petersburg, Russia) with positive polarity, equipped with a special geometry quartz capillary similar to amino acid (pore inner diameter 50 microns, total/effective length 75/65 cm) and the specialized software Elforun^®^ (S-Pb, St. Petersburg, Russia). The preparation of samples for research consisted of the microbial cells’ precipitation using an Eppendorf 5810R centrifuge (Eppendorf, Hamburg, Germany) at 7500 rpm for 10 min. After that, the supernatants were placed in Amicon Ultra-15 centrifuge concentrators (Merck, Darmstadt, Germany) with a 3 kDa pore diameter cellulose membrane and centrifuged for 15 min at 7500 rpm for complete purification. Separation, identification, and determination of the concentrations of water-soluble vitamins thiamine (B1), riboflavin (B2), pantothenic acid (B3), pyridoxine (B6), and folic acid (Bc) were carried out by capillary zone electrophoresis (CSE), and that of the vitamin nicotinamide (B5) by micellar electrokinetic chromatography (MEKC) according to the practical guidelines for the use of “Kapel” capillary electrophoresis systems [30]. Detection conditions for vitamins B1, B2, B3, B6, and Bc: capillary: quartz, 50 microns, 75 cm; voltage + 25 kV; wavelength at the beginning of the analysis 200 nm, after the detection peak B1 267 nm, after the peak B2 and before the end of the analysis 200 nm; temperature 30 °C; pressure at the beginning of the analysis 0 mbar, after the peak B6-30 mbar until the end of the analysis. Detection conditions for vitamin B5: capillary: quartz, 50 microns, 75 cm; voltage +25 kV; wavelength 200 nm; temperature 40 °C; pressure 0 mbar. The standards (HPLC grade) of all B vitamins for calibration were purchased from Lumex, Ltd. All samples were tested in triplicate.

### 2.6. Statistical Analysis

All the measurements were performed in triplicate, and the results were expressed as the mean with standard deviation (±SD). The data were analyzed by an ANOVA test with Tukey’s multiple comparison using MathLab (MathWorks, Natick, MA, USA) software. Statistical significance was determined at *p* = 0.05.

## 3. Results

### 3.1. Stress Tolerance Ability

#### 3.1.1. Tolerance of *Bifidobacterium* and *Lactobacillus* Strains to Simulated Gastric Juices

The standard conditions for incubation in real or simulated gastric juices have not been fully developed yet [31]. However, it is known that, depending on the consumed food and beverages, incubation time in the stomach can vary from 15 to 194 min [32]. Therefore, to simulate the presence of lactobacilli and bifidobacteria in the stomach, an incubation period of 90 min was determined. As the death rate of bifidobacteria will be greater than that of lactobacilli, the measurements for the strains of *Bifidobacterium* genus were carried out more frequently. In this case, only the final results of the test (after 90 min) are applicable for intergenera comparison.

The results of the research on the survival of six bifidobacteria strains and 10 lactobacilli strains during the incubation in simulated gastric juices at pH 2.0 for 1.5 h are shown in Table 2 and Table 3, respectively. Each tested strain of bifidobacteria demonstrated a considerable decrease in the viable cell count during the first 30 min of incubation. The strains of *B. adolescentis* VKPM AC-1662, *B. pseudolongum* subsp. *pseudolongum* VKPM AC-1785, and *B. breve* VKPM AC-1911 showed the highest tolerance to the effects of simulated gastric juices. The marked decrease in the population was observed in the *B. longum* subsp. *infantis* VKPM A C-1912, *B. longum* subsp. *longum* VKPM AC-1665, and *B. bifidum* VKPM AC-1779 strains. The strain *B. bifidum* VKPM AC-1779 was characterized by the decay of viable cells after 40 min of exposure.

All the studied lactobacillus strains also demonstrated a decrease in the viable cell count under the model conditions of simulated gastric juices (Table 3). The strain *L. sakei* VKPM B-8936 demonstrated the greatest resistance to low pH values, while *L. salivarius* VKPM B-2214 had the lowest. Total inactivation of the lactobacillus population was not observed for any strain after 1.5 h of incubation.

#### 3.1.2. Tolerance of *Bifidobacterium* and *Lactobacillus* Strains to the Simulated Duodenal Environment

The time of food transit via the small intestine is usually 1 to 4 h, which gives a total transit time in the upper gastrointestinal tract of 3 to 8 h [33]. Therefore, in order to simulate the passing of lactobacilli and bifidobacteria in the intestine, the incubation period of 3 h was established. The viability of lactobacilli after 3 h in the duodenum-simulating conditions was comparatively higher than in bifidobacteria (Table 4 and Table 5). At the same time, the survival rate of both lactobacilli and bifidobacteria was strain-specific and differed between the cultures.

The strain *B. adolescentis* VKPM AC-1662 had the highest survival rate, almost equal to 100%. The greatest sensitivity to the conditions of the intestinal environment was demonstrated by the bifidobacteria *B. pseudolongum* subsp. *pseudolongum* VKPM AC-1785 and *B. breve* VKPM AC-1911 strains, which had the survival percentage of 80% after 3 h of incubation.

Among lactobacilli, the lowest decrease in population was observed in the *L. plantarum* VKPM B-11007 and *L. acidophilus* VKPM B-2213 strains. The survival rate of the *L. salivarius* VKPM B-2214 strain was the lowest.

#### 3.1.3. Tolerance of *Bifidobacterium* and *Lactobacillus* Strains to Digestive Enzymes

To clarify the effect of enzymes as principal components of digestive juices, the strains were tested separately for pepsin and pancreatin action. A study of the effect of the proteolytic enzymes, such as pepsin and pancreatin, on the survival of bifidobacteria showed that they had an equally strong effect on every considered strain. Total death of microorganisms occurred after 1 h of incubation The results of the research on the sensitivity of lactobacilli to pepsin and pancreatin are shown in Table 6 and Table 7.

The inactivation effect of pepsin on the lactobacilli viability was higher than that of pancreatin. If the survival rate of lactobacilli after 3 h of incubation with pancreatin was about 90% for all the cultures, the sensitivity to pepsin greatly varied between the strains, reaching 60% in some cases. Thus, the strain *L. paracasei* subsp. *paracasei* VKPM B-4079 demonstrated the lowest tolerance to pepsin, and *L. acidophilus* VKPM B-2213 the greatest.

#### 3.1.4. Tolerance of *Bifidobacterium* and *Lactobacillus* Strains to Bile

The results of the study on bile’s effect on the viability of bifidobacteria and lactic acid bacteria strains are shown in Table 8 and Table 9. Almost every strain considered showed a decrease in the population of viable cells. The most sensitive to the presence of bile in the medium was the *B. breve* VKPM AC-1911 strain (the survival rate was 70%). The *B. bifidum* VKPM AC-1779 strain was characterized by the greatest resistance to the effects of bile; the survival rate hardly changed (about 99%).

Among the *Lactobacillus* strains considered, the *L. acidophilus* VKPM B-2213 strain demonstrated the greatest sensitivity to bile, and *L. acidophilus* VKPM B-2105 the lowest.

### 3.2. Autoaggregation Assays of Bifidobacterium and Lactobacillus Strains

The results of the study on the autoaggregation of bifidobacteria strains are shown in Table 10. The highest percentage of autoaggregation was demonstrated by the strain *B. adolescentis* VKPM AC-1662: 69% after 24 h of incubation. *B. longum* subsp. *longum* VKPM AC-1665 aggregation was a little lower. The other strains demonstrated low autoaggregation ability.

The results of the *Lactobacillus* strains autoaggregation test are shown in Table 11. The greatest autoaggregation was observed in the *L. salivarius* VKPM B-2214 strain: 56% after 24 h of incubation. Also, a high autoaggregation ability was observed in the *L. acidophilus* VKPM B-6551, *L. acidophilus* VKPM B-2213, and *L. rhamnosus* VKPM B-8238 strains. The aggregation percentages of the other lactobacilli strains considered were lower.

Thus, the strains *B. adolescentis* VKPM AC-1662, *B. longum* subsp. *longum* ВКПМ AC-1665, *L. salivarius* VKPM B-2214, *L. acidophilus* VKPM B-6551, *L. acidophilus* VKPM B-2213, and *L. rhamnosus* VKPM B-8238 have a high potential ability to attach to epithelial cells and mucosal surfaces.

### 3.3. The Study of Lactic Acid Bacteria’s and Bifidobacteria’s Ability to Accumulate B Vitamins Extracellularly

While studying the ability of lactic acid bacteria and bifidobacteria to produce B vitamins, it was assumed that, if a specific vitamin was found in the supernatant, the considered strain was able not only to synthesize, but also to excrete vitamins simultaneously with their accumulation in the medium. The results of the studies are presented in Table 12 and Table 13.

Based on the results, each bifidobacteria culture considered consumed nicotinamide (its concentration was higher in the medium before fermentation). The riboflavin that was detected in the nutrient medium was consumed by all cultures except *B. adolescentis* AC1662 and *B. longum* subsp. *infantis* AC-1912, which excrete it in the medium, such that an increase in concentration was observed. The ability to accumulate pyridoxine extracellularly was found in *B. adolescentis* VKPM AC-1662, *B. pseudolongum* subsp. *pseudolongum* VKPM AC-1785, *B. longum* subsp. *infantis* VKPM AC-1912, and *B. longum* VKPM AC-1665, among which the maximum amount was detected for the last one. Thiamine, pantothenic acid, and folic acid were not detected in the supernatant of the fermentation broths.

Each *Lactobacillus* strain also consumed nicotinamide. The ability to produce thiamine in the medium was found in *L. casei* VKPM B-2873 and *L. rhamnosus* VKPM B-8238; pantothenic acid in *L. acidophilus* VKPM B-2105, *L. salivarius* VKPM B-2214, *L. casei* VKPM B-2873, and *L. rhamnosus* VKPM B-8238; and pyridoxine in *L. acidophilus* VKPM B-2105, *L. plantarum* VKPM B-11007, *L. acidophilus* VKPM B-6551, *L. casei* VKPM B-2873, *L. rhamnosus* VKPM B-8238, and *L. sakei* VKPM B-8936. Folic acid was detected in the fermentation broths of *L. salivarius* VKPM B-2214 and *L. acidophilus* VKPM B-6553.

Among the *Lactobacillus* strains considered, *L. salivarius* VKPM B-2214 is of the greatest interest since the pantothenic acid concentration reached 138.600 mg L^−1^ in the fermentation broth.

## 4. Discussion

Carrying out proper in vitro studies to screen potential probiotic strains is a necessary first step before in vivo experiments to establish the possible health benefits of probiotics [34]. An important evaluation criterion during the selection is stress tolerance to passing through the digestive tract, which characterizes the resistance of the probiotics to acid and bile, as well as the production of antimicrobial compounds and the ability to aggregate with intestinal cells [34,35]. The acidic gastric juices and bile salts are the main obstacle to the survival of probiotic bacteria and reaching the distal part of the intestine [36].

The results obtained in our research show that the stress tolerance of bifidobacteria and lactobacilli to the effects of low pH, proteolytic enzymes, and bile varies greatly, both between species and between strains of the given species. The effect of artificial gastric juices on bifidobacteria strains led to the death of *B. longum* subsp. *infantis* VKPM AC-1912, *B. longum* subsp. *longum* VKPM AC-1665, and *B. bifidum* VKPM AC-1779. Similar results were reported by Charteris et al. [37], who showed a lack of resistance to simulated gastric conditions (pH 2.0 for 90 min) in the *B. bifidum*, *B. animalis*, *B. infantis*, *B. breve*, and *B. adolescentis* strains. Strong differences in resistance to low pH between strains within the species *B. longum* were observed by Izquierdo et al. [38]. In our study, the *B. adolescentis* VKPM AC-1662 strain, which was isolated from the human intestine, was characterized by the greatest acid resistance. The effect of the model duodenum conditions on the bifidobacteria survival was less harmful, and the survival rate for all strains was above 80%. The strain *B. adolescentis* VKPM AC-1662 also showed the highest stability.

When comparing the tolerance of the tested lactobacilli and bifidobacteria to the gastrointestinal tract model conditions, the higher resistance of the former was anticipated. They were resistant to both simulated gastric juices and bile. Acid resistance is a characteristic of most lactobacillus species due to their ability to control intracellular pH [39]. Some of them (e.g., *Lactobacillus casei*, *Lactobacillus reuteri*, *Lactobacillus vaginalis*, *Lactobacillus fermentum*, and *Lactobacillus casei*) may be members of the gastric microbial community [40]. In our research, the strain *L. sakei* VKPM B-8936 showed the greatest resistance to low pH. Similar results were reported by Song et al. [41], who showed the acid resistance of the *Lactobacillus* sp. NN 8829, *L. casei* MB3, *L. sakei* MA9, and *L. sakei* CH8 strains under incubation conditions at pH 2.5. Thus, the obtained data are generally consistent with the results presented earlier. On the other hand, taking into account the confirmed strain-specific tolerance, the accumulation of information on new strains is theoretically and practically valuable. The experiment dedicated to the treatment of bifidobacteria populations separately with pepsin and pancreatin led to the death of all tested strains. Lactobacilli were more resistant to the effects of proteolytic enzymes. The greater sensitivity of lactobacilli was marked in the experiments with pepsin.

The bacterial cell aggregation of probiotic strains is often associated with potential ability to adhere to the intestinal epithelium and mucous membranes [17]. In most cases, bacterial autoaggregation is mediated by homotypic interactions between surface proteins [42]. The tested cultures of lactic acid bacteria and bifidobacteria demonstrated some degree of autoaggregation, which differed greatly between the strains. The strains *B. adolescentis* VKPM AC-1662, *B. longum* subsp. *longum* VKPM AC-1665, and *B. bifidum* VKPM AC-1779, isolated from the intestines of adults, showed strong autoaggregation ability. Among lactobacilli, the highest percentage of autoaggregation was observed in *L. salivarius* VKPM B-2214, isolated from human saliva. It was also characterized by the strong dependence of bacterial cells’ aggregation on exposure time, which was greatest after 24 h. It was reported previously that *L. acidophilus* is characterized by high values of autoaggregation [43]. In our study, high autoaggregation ability was also observed in *L. acidophilus* VKPM B-6551, *L. acidophilus* VKPM B-2213, and *L. rhamnosus* VKPM B-8238.

The reference probiotic bacteria, e.g., *L. rhamnosus* GG or *L. casei* Shirota, can be applied in research on probiotic properties to improve the reliability of the data comparison [44,45,46]. The lack of a reference probiotic can be considered a limitation of our study. However, it should be noted that the standards were not used in the numerous works mentioned above. The FAO/WHO guidelines [47] also do not establish strict requirements. Moreover, discrepancies in the data on the stress tolerance of the same reference strain and in the same tests were noted between studies.

The analysis of the obtained results on lactic acid bacteria’s and bifidobacteria’s ability to extracellularly production of water-soluble B vitamins showed that the consumption of some vitamins detected in the nutrient medium before fermentation and the excretion of others into fermentation broths occurred. So, riboflavin was not consumed by *B. adolescentis* VKPM AC-1662 and *B. longum* subsp. *infantis* VKPM AC-1912. Hou et al. [48] showed that fermentation of soy milk with *Bifidobacterium longum* B6 and *B. infantis* CCRC 14633 cultures increases the content of riboflavin. Pyridoxine accumulation was observed in the strains of *B. adolescentis* VKPM AC-1662, *B. pseudolongum* subsp. *pseudolongum* VKPM AC-1785, *B. longum* subsp. *infantis* VKPM AC-1912, and *B. longum* VKPM AC-1665. The obtained results are consistent with the data of Deguchi, Morishita and Mutai [49], who linked the ability to synthesize certain vitamins with the species and strain. The extracellular synthesis of folic acid was demonstrated in the studies of Deguchi, Morishita and Mutai [50] and Pompei et al. [50], but it was not detected for the considered strains. Although *Bifidobacterium bifidum* and *Bifidobacterium longum* subsp. *infantis* are classified as species/strains with a high level of biosynthetic properties of folic acid [50].

The B vitamins’ extracellular synthesis by lactobacilli was poor, especially when modern techniques such as capillary electrophoresis were applied. In our study, the ability to excrete thiamine, folic acid, and pantothenic acid was considered for some strains of lactobacilli. *L. salivarius* VKPM B-2214 is of interest due to its ability to produce up to 138.600 mg L^−1^ of pantothenic acid.

Thus, *B. adolescentis* VKPM AC-1662 can be considered the best candidate for probiotics, since it is characterized by the greatest stress tolerance and a high autoaggregation percentage, as well as the ability to accumulate riboflavin and pyridoxine extracellularly. The application of folate-producing bifidobacteria has already been shown in rats [51] and humans [52]. On the other hand, data on the correlation of the synthesis of B vitamins in vitro and in the intestine are poor; additional research is required, which may involve significant methodological difficulties. Among the lactic acid bacteria, the greatest tolerance to low pH values was demonstrated by *L. sakei* VKPM B-8936; *L. plantarum* VKPM B-11007 showed tolerance to duodenal conditions, *L. acidophilus* VKPM B-2213 to pepsin, and *L. salivarius* VKPM B-2214 to pancreatin. The highest autoaggregation percentage was observed in *L. salivarius* VKPM B-2214, which also accumulated the most pantothenic acid of the studied strains. *L. casei* VKPM B-2873 and *L. rhamnosus* VKPM B-8238 was also characterized by the accumulation of thiamine and pyridoxine in addition to pantothenic acid. These strains of lactobacilli have the potential to be used as starter cultures in functional product manufacturing.

## 5. Conclusions

In this research, for the first time, the combined assessment of the probiotic and vitamin-producing potential of various bifidobacteria and lactobacilli species and strains was carried out in order to select potential probiotics with high stress tolerance and the ability to synthesize B vitamins. The selected strains with the best characteristics, such as *L. salivarius* VKPM B-2214, *L. casei* VKPM B-2873, and *L. rhamnosus* VKPM B-8238, can be used as potential starter cultures to increase B vitamins in a number of probiotic foods. *B. adolescentis* VKPM AC-1662 can be introduced in the diet as a probiotic, potentially capable to synthetize B vitamins in vivo in the intestine. The conducted studies of lactobacilli and bifidobacteria strains, devoted to the ability to extracellularly produce B vitamins in combination with the assessment of probiotic properties, can be considered the first stage in their introduction for commercial use. It should be pointed out that studies on B vitamin accumulation in the bacterial biomass, as well as their production in other nutrient media, have not been carried out in this work. Therefore, research into both the effect of various cultivation conditions on the accumulation of B vitamins (in order to increase their concentrations), and the fermentation of the strains in other nutrient media (including milk or plant substrates) should be undertaken in the future.

## Figures and Tables

**Table 1 microorganisms-10-00470-t001:** The origins of the organisms used in the study.

Species	Strains	Origin
*Bifidobacterium adolescentis*	VKPM AC-1662	Human
*Bifidobacterium pseudolongum* subsp. *pseudolongum*	VKPM AC-1785	Pig
*Bifidobacterium longum* subsp. *infantis*	VKPM AC-1912	Human
*Bifidobacterium breve*	VKPM AC-1911	Human
*Bifidobacterium longum* subsp. *longum*	VKPM AC-1665	Human
*Bifidobacterium bifidum*	VKPM AC-1779	Human
*Lactobacillus acidophilus*	VKPM B-2105	Fermented dairy product
*Lactobacillus plantarum*	VKPM B-11007	Lactobacterin (medication)
*Lactobacillus acidophilus*	VKPM B-6551	Human
*Lactobacillus acidophilus*	VKPM B-2213	Human
*Lactobacillus salivarius*	VKPM B-2214	Human
*Lactobacillus casei*	VKPM B-2873	Human
*Lactobacillus acidophilus*	VKPM B-6553	Human
*Lactobacillus rhamnosus*	VKPM B-8238	Human
*Lactobacillus sakei*	VKPM B-8936	Italian dry-cured sausage
*Lactobacillus paracasei* subsp. *paracasei*	VKPM B-4079	Fermented sugar beet tops

**Table 2 microorganisms-10-00470-t002:** Tolerance of *Bifidobacterium* strains to simulated gastric juices (pH 2.0).

Strain	Survival Percentage (% ± SD)			
	Time (min)			
	20	40	60	90
*B. adolescentis* VKPM AC-1662	95 ± 0.4 ^a^	81 ± 0.9 ^a^	71 ± 0.2 ^a^	67 ± 0.7 ^a^
*B. pseudolongum* subsp. *pseudolongum* VKPM AC-1785	88 ± 0.9 ^b^	86 ± 1.1 ^b^	63 ± 0.4 ^b^	65 ± 0.8 ^b^
*B. longum* subsp. *infantis* VKPM AC-1912	75 ± 0.2 ^c^	54 ± 0.9 ^c^	0 ^c^	0 ^c^
*B. breve* VKPM AC-1911	81 ± 1.1 ^d^	60 ± 0.6 ^d^	60 ± 0.2 ^d^	60 ± 0.8 ^d^
*B. longum* subsp. *longum* VKPM AC-1665	80 ± 1.2 ^d^	50 ± 1.2 ^e^	0 ^c^	0 ^c^
*B. bifidum* VKPM AC-1779	63 ± 0.5 ^e^	0 ^f^	0 ^c^	0 ^c^

Values are the mean ± SD of three determinations of three replicates. The superscripts indicate statistically significant differences (*p* < 0.05) within each time comparison.

**Table 3 microorganisms-10-00470-t003:** Tolerance of *Lactobacillus* strains to simulated gastric juices (pH 2.0).

Strain	Survival Percentage (% ± SD)		
	Time (min)		
	30	60	90
*L. acidophilus* VKPM B-2105	97 ± 0.7 ^a^	84 ± 1.2 ^a^	81 ± 0.7 ^a^
*L. plantarum* VKPM B-11007	99 ± 0.3 ^b^	84 ± 0.5 ^a^	71 ± 0.8 ^b^
*L. acidophilus* VKPM B-6551	96 ± 1.4 ^a^	84 ± 0.6 ^b^	82 ± 0.6 ^c^
*L. acidophilus* VKPM B-2213	99 ± 0.5 ^b^	86 ± 0.1 ^c^	74 ± 2.0 ^d^
*L. salivarius* VKPM B-2214	89 ± 0.8 ^c^	89 ± 0.5 ^d^	72 ± 0.4 ^e^
*L. casei* VKPM B-2873	89 ± 0.9 ^c^	89 ± 1.5 ^e^	72 ± 0.8 ^f^
*L. acidophilus* VKPM B-6553	90 ± 0.8 ^c^	88 ± 1.4 ^e^	87 ± 0.3 ^g^
*L. rhamnosus* VKPM B-8238	97 ± 1.1 ^a,b^	93 ± 0.9 ^f^	81 ± 1.3 ^a,c^
*L. sakei* VKPM B-8936	98 ± 0.7 ^a,b^	94 ± 0.4 ^f^	93 ± 0.3 ^h^
*L. paracasei* subsp. *paracasei* VKPM B-4079	85 ± 1.6 ^d^	81 ± 1.0 ^f^	78 ± 0.6 ^i^

Values are the mean ± SD of three determinations of three replicates. The superscripts indicate statistically significant differences (*p* < 0.05) within each time comparison.

**Table 4 microorganisms-10-00470-t004:** Tolerance of *Bifidobacterium* strains to simulated duodenal environment (pH 6.0).

Strain	Survival Percentage (% ± SD)		
	Time (min)		
	60	120	180
*B. adolescentis* VKPM AC-1662	100 ± 0.02 ^a^	100 ± 0.03 ^a^	98 ± 0.7 ^a^
*B. pseudolongum* subsp. *pseudolongum* VKPM AC-1785	100 ± 0.03 ^a^	92 ± 1.6 ^b^	80 ± 2.1 ^b^
*B. longum* subsp. *infantis* VKPM AC-1912	96 ± 1.7 ^b^	86 ± 0.4 ^c^	86 ± 1.5 ^c^
*B. breve* VKPM AC-1911	89 ± 1.0 ^c^	92 ± 0.5 ^b^	80 ± 0.6 ^b^
*B. longum* subsp. *longum* VKPM AC-1665	87 ± 0.8 ^d^	87 ± 0.5 ^c^	87 ± 0.2 ^c^
*B. bifidum* VKPM AC-1779	89 ± 1.1 ^c^	87 ± 0.9 ^c^	87 ± 1.3 ^c^

Values are the mean ± SD of three determinations of three replicates. The superscripts indicate statistically significant differences (*p* < 0.05) within each time comparison.

**Table 5 microorganisms-10-00470-t005:** Tolerance of *Lactobacillus* strains to simulated duodenal environment (pH 6.0).

Strain	Survival Percentage (%±SD)		
	Time (min)		
	60	120	180
*L. acidophilus* VKPM B-2105	91 ± 1.7 ^a^	90 ± 0.9 ^a^	89 ± 0.6 ^a^
*L. plantarum* VKPM B-11007	100 ± 0.03 ^b^	99 ± 0.4 ^b^	99 ± 0.6 ^b^
*L. acidophilus* VKPM B-6551	100 ± 0.04 ^b^	86 ± 1.1 ^c^	85 ± 0.9 ^c^
*L. acidophilus* VKPM B-2213	100 ± 0.04 ^b^	100 ± 0.03 ^b^	99 ± 0.4 ^b^
*L. salivarius* VKPM B-2214	95 ± 0.4 ^c^	84 ± 1.6 ^d^	84 ± 1.9 ^c^
*L. casei* VKPM B-2873	98 ± 0.6 ^d^	94 ± 0.8 ^e^	89 ± 1.3 ^a^
*L. acidophilus* VKPM B-6553	96 ± 0.9 ^e^	92 ± 1.2 ^f^	86 ± 0.7 ^c^
*L. rhamnosus* VKPM B-8238	99 ± 0.3 ^b,d^	96 ± 1.4 ^g^	92 ± 0.8 ^d^
*L. sakei* VKPM B-8936	99 ± 0.8 ^b,d^	97 ± 0.2 ^g^	89 ± 0.3 ^a^
*L. paracasei* subsp. *paracasei* VKPM B-4079	99 ± 0.6 ^b,d^	97 ± 0.9 ^g^	95 ± 1.1 ^e^

Values are the mean ± SD of three determinations of three replicates. The superscripts indicate statistically significant differences (*p* < 0.05) within each time comparison.

**Table 6 microorganisms-10-00470-t006:** Tolerance of *Lactobacillus* strains to pepsin.

Strain	Survival Percentage (% ± SD)		
	Time (min)		
	60	120	180
*L. acidophilus* VKPM B-2105	96 ± 1.5 ^a^	80 ± 1.3 ^a^	80 ± 0.6 ^a^
*L. plantarum* VKPM B-11007	98 ± 0.4 ^b^	83 ± 1.2 ^b^	72 ± 0.9 ^b^
*L. acidophilus* VKPM B-6551	92 ± 0.5 ^c^	85 ± 0.8 ^c^	85 ± 0.2 ^c^
*L. acidophilus* VKPM B-2213	98 ± 0.03 ^a,b^	92 ± 0.6 ^d^	91 ± 0.7 ^d^
*L. salivarius* VKPM B-2214	94 ± 1.9 ^d^	88 ± 1.0 ^e^	84 ± 0.6 ^e^
*L. casei* VKPM B-2873	92 ± 0.5 ^c,d^	84 ± 0.8 ^b,c^	81 ± 1.1 ^a^
*L. acidophilus* VKPM B-6553	96 ± 0.9 ^a^	91 ± 0.6 ^d^	83 ± 1.3 ^e^
*L. rhamnosus* VKPM B-8238	87 ± 0.7 ^e^	74 ± 1.5 ^f^	71 ± 0.8 ^b^
*L. sakei* VKPM B-8936	90 ± 1.2 ^f^	84 ± 0.7 ^c^	73 ± 0.1 ^b^
*L. paracasei* subsp. *paracasei* VKPM B-4079	73 ± 0.2 ^g^	67 ± 1.6 ^g^	58 ± 1.0 ^f^

Values are the mean ± SD of three determinations of three replicates. The superscripts indicate statistically significant differences (*p* < 0.05) within each time comparison.

**Table 7 microorganisms-10-00470-t007:** Tolerance of *Lactobacillus* strains to pancreatin.

Strain	Survival Percentage (%±SD)		
	Time (min)		
	60	120	180
*L. acidophilus* VKPM B-2105	100 ± 0.03 ^a^	92 ± 1.0 ^a^	91 ± 0.8 ^a^
*L. plantarum* VKPM B-11007	97 ± 0.7 ^b^	92 ± 0.9 ^a^	91 ± 0.2 ^a^
*L. acidophilus* VKPM B-6551	100 ± 0.1 ^a^	95 ± 0.5 ^b^	94 ± 0.6 ^b^
*L. acidophilus* VKPM B-2213	98 ± 0.7 ^c^	95 ± 0.9 ^b,c^	93 ± 0.5 ^b^
*L. salivarius* VKPM B-2214	99 ± 0.1 ^a^	98 ± 0.4 ^d^	98 ± 0.4 ^c^
*L. casei* VKPM B-2873	98 ± 0.6 ^c^	96 ± 0.3 ^c^	93 ± 0.9 ^b^
*L. acidophilus* VKPM B-6553	98 ± 0.3 ^b,c^	98 ± 0.5 ^d^	97 ± 1.2 ^c,d^
*L. rhamnosus* VKPM B-8238	100 ± 0.2 ^a^	96 ± 1.4 ^b,c^	96 ± 0.9 ^d^
*L. sakei* VKPM B-8936	95 ± 1.0 ^d^	94 ± 0.6 ^b^	88 ± 1.4 ^e^
*L. paracasei* subsp. *paracasei* VKPM B-4079	96 ± 0.6 ^e^	92 ± 0.9 ^a^	90 ± 1.5 ^f^

Values are the mean ± SD of three determinations of three replicates. The superscripts indicate statistically significant differences (*p* < 0.05) within each time comparison.

**Table 8 microorganisms-10-00470-t008:** Tolerance of *Bifidobacterium* strains to bile.

Strain	Survival Percentage (% ± SD)		
	Time (min)		
	60	120	180
*B. adolescentis* VKPM AC-1662	89 ± 0.6 ^a^	89 ± 1.3 ^a^	89 ± 1.5 ^a^
*B. pseudolongum* subsp. *pseudolongum* VKPM AC-1785	99 ± 0.1 ^b^	89 ± 0.8 ^a^	88 ± 1.8 ^a^
*B. longum* subsp. *infantis* VKPM AC-1912	94 ± 0.4 ^c^	88 ± 1.0 ^a^	87 ± 1.1 ^a^
*B. breve* VKPM AC-1911	89 ± 0.8 ^a^	79 ± 1.8 ^b^	70 ± 0.5 ^b^
*B. longum* subsp. *longum* VKPM AC-1665	98 ± 0.4 ^d^	89 ± 1.1 ^a^	80 ± 1.7 ^c^
*B. bifidum* VKPM AC-1779	99 ± 0.2 ^b^	99 ± 0.1 ^c^	99 ± 0.04 ^d^

Values are the mean ± SD of three determinations of three replicates. The superscripts indicate statistically significant differences (*p* < 0.05) within each time comparison.

**Table 9 microorganisms-10-00470-t009:** Tolerance of *Lactobacillus* strains to bile.

Strain	Survival Percentage (% ± SD)		
	Time (min)		
	60	120	180
*L. acidophilus* VKPM B-2105	98 ± 0.2 ^a^	98 ± 0.6 ^a^	98 ± 0.7 ^a^
*L. plantarum* VKPM B-11007	90 ± 2.0 ^b^	88 ± 1.2 ^b^	86 ± 0.6 ^b^
*L. acidophilus* VKPM B-6551	88 ± 1.6 ^c^	81 ± 0.7 ^c^	79 ± 1.9 ^c^
*L. acidophilus* VKPM B-2213	88 ± 0.6 ^c,d^	74 ± 1.0 ^d^	70 ± 1.0 ^d^
*L. salivarius* VKPM B-2214	97 ± 0.5 ^a^	90 ± 0.8 ^e^	76 ± 0.5 ^e^
*L. casei* VKPM B-2873	95 ± 0.1 ^e^	85 ± 0.9 ^f^	83 ± 1.2 ^f^
*L. acidophilus* VKPM B-6553	88 ± 1.3 ^c^	81 ± 0.3 ^c^	78 ± 1.5 ^c^
*L. rhamnosus* VKPM B-8238	86 ± 0.8 ^d^	76 ± 0.3 ^g^	72 ± 0.4 ^g^
*L. sakei* VKPM B-8936	95 ± 0.2 ^e^	85 ± 0.9 ^f^	77 ± 1.0 ^c,e^
*L. paracasei* subsp. *paracasei* VKPM B-4079	88 ± 1.7 ^c,d^	82 ± 0.4 ^c^	80 ± 0.6 ^c^

Values are the mean ± SD of three determinations of three replicates. The superscripts indicate statistically significant differences (*p* < 0.05) within each time comparison.

**Table 10 microorganisms-10-00470-t010:** Autoaggregation percentage of *Bifidobacterium* strains.

Strains	Time			
	3 h	6 h	12 h	24 h
*B. adolescentis* VKPM AC-1662	15 ± 0.5 ^a^	48 ± 0.8 ^a^	53 ± 0.8 ^a^	69 ± 0.8 ^a^
*B. pseudolongum* subsp. *pseudolongum* VKPM AC-1785	9 ± 0.8 ^b^	12 ± 0.4 ^b^	14 ± 0.3 ^b^	13 ± 0.3 ^b^
*B. longum* subsp. *infantis* VKPM AC-1912	12 ± 0.4 ^c^	15 ± 0.5 ^c^	13 ± 0.2 ^b^	11 ± 0.2 ^c^
*B. breve* VKPM AC-1911	12 ± 0.4 ^c^	32 ± 0.6 ^d^	30 ± 0.5 ^c^	31 ± 0.2 ^d^
*B. longum* subsp. *longum* VKPM AC-1665	20 ± 0.2 ^d^	56 ± 0.2 ^e^	59 ± 0.6 ^d^	67 ± 1.0 ^e^
*B. bifidum* VKPM AC-1779	34 ± 0.9 ^e^	47 ± 0.6 ^a^	55 ± 0.8 ^e^	57 ± 0.9 ^f^

Values are the mean ± SD of three determinations of three replicates. The superscripts indicate statistically significant differences (*p* < 0.05) within each time comparison.

**Table 11 microorganisms-10-00470-t011:** Autoaggregation percentage of *Lactobacillus* strains.

Strains	Time			
	3 h	6 h	12 h	24 h
*L. acidophilus* VKPM B-2105	10 ± 0.4 ^a^	28 ± 0.5 ^a^	40 ± 0.9 ^a^	47 ± 1.0 ^a^
*L. plantarum* VKPM B-11007	9 ± 0.2 ^b^	9 ± 0.2 ^b^	7 ± 0.6 ^b^	8 ± 0.5 ^b^
*L. acidophilus* VKPM B-6551	25 ± 0.3 ^c^	50 ± 0.8 ^c^	52 ± 0.7 ^c^	55 ± 0.8 ^c^
*L. acidophilus* VKPM B-2213	12 ± 0.3 ^d^	49 ± 0.2 ^d^	50 ± 0.6 ^d^	53 ± 0.6 ^d^
*L. salivarius* VKPM B-2214	14 ± 0.2 ^e^	18 ± 0.1 ^e^	26 ± 0.7 ^e^	56 ± 1.1 ^c^
*L. casei* VKPM B-2873	6 ± 0.1 ^f^	13 ± 0.3 ^f^	12 ± 0.3 ^f^	32 ± 0.7 ^e^
*L. acidophilus* VKPM B-6553	5 ± 0.2 ^g^	8 ± 0.1 ^g^	8 ± 0.3 ^g^	25 ± 0.3 ^f^
*L. rhamnosus* VKPM B-8238	16 ± 0.3 ^h^	44 ± 0.8 ^h^	46 ± 0.8 ^h^	50 ± 0.7 ^g^
*L. sakei* VKPM B-8936	11 ± 0.1 ^i^	10 ± 0.2 ^i^	10 ± 0.2 ^i^	12 ± 0.5 ^h^
*L. paracasei* subsp. *paracasei* VKPM B-4079	5 ± 0.2 ^g^	8 ± 0.1 ^g^	5 ± 0.2 ^j^	7 ± 0.2 ^b^

Values are the mean ± SD of three determinations of three replicates. The superscripts indicate statistically significant differences (*p* < 0.05) within each time comparison.

**Table 12 microorganisms-10-00470-t012:** Concentration of B vitamins in bifidobacteria fermentation broths.

Strain	Vitamin Concentrations, mg L^−1^					
	Thiamine (B1)	Riboflavin (B2)	Pantothenic Acid (B3)	Nicotinamide (B5)	Pyridoxine (B6)	Folic Acid (Bc)
*B. adolescentis* VKPM AC-1662	—	2.180 ± 0.021	—	1.462 ± 0.040	7.974 ± 0.040	—
*B. pseudolongum* VKPM AC-1785	—	—	—	1.554 ± 0.020	20.680 ± 0.080	—
*B. longum subsp. infantis* VKPM AC-1912	—	2.009 ± 0.011	—	44.130 ± 0.100	0.988 ± 0.010	—
*B. breve* VKPM AC-1911	—	0.503 ± 0.012	—	45.730 ± 0.110	—	—
*B. longum* VKPM AC-1665	—	—	—	12.010 ± 0.021	26.060 ± 0.030	—
*B. bifidum* VKPM AC-1779	—	0.613 ± 0.031	—	0.304 ± 0.060	—	—
medium before fermentation	—	1.191 ± 0.011	—	52.970 ± 0.091	—	—

Values are the mean ± SD of three determinations of three replicates. The concentrations of each B vitamin differed significantly between the strains (*p* < 0.05).

**Table 13 microorganisms-10-00470-t013:** Concentration of B vitamins in lactic acid bacteria fermentation broths.

Strain	Vitamin Concentrations, mg L^−1^					
	Thiamine (B1)	Riboflavin (B2)	Pantothenic Acid (B3)	Nicotinamide (B5)	Pyridoxine (B6)	Folic Acid (Bc)
*L. acidophilus* VKPM B-2105	—	0.917 ± 0.010	10.220 ± 0.032	43.640 ± 0.050	4.096 ± 0.020	—
*L. plantarum* VKPM B-11007	—	—	—	46.650 ± 0.020	3.453 ± 0.010	—
*L. acidophilus* VKPM B-6551	—	—	—	0.671 ± 0.010	1.100 ± 0.010	—
*L. acidophilus* VKPM B-2213	—	0.171 ± 0.010	—	49.920 ± 0.011	—	—
*L. salivarius* VKPM B-2214	—	—	138.600 ± 0.021	38.130 ± 0.030	—	1.500 ± 0.010
*L. casei* VKPM B-2873	19.540 ± 0.023	—	3.966 ± 0.020	—	11.400 ± 0.030	—
*L. acidophilus* VKPM B-6553	—	0.472 ± 0.020	—	43.030 ± 0.030	—	2.944 ± 0.020
*L. rhamnosus* VKPM B-8238	14.72 ± 0.010	—	52.590 ± 0.100	27.690 ± 0.010	2.098 ± 0.010	—
*L. sakei* VKPM B-8936	—	—	—	5.111 ± 0.010	6.824 ± 0.010	—
*L. paracasei* subsp. *paracasei* VKPM B-4079	—	—	—	13.880 ± 0.012	—	—
medium before fermentation	—	—	—	61.370 ± 0.010	—	—

Values are the mean ± SD of three determinations of three replicates. The concentrations of each B vitamin differed significantly between the strains (*p* < 0.05).

## Data Availability

Data is contained within the article.

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
