# Peer review of "The Combination of In Vitro Assessment of Stress Tolerance Ability, Autoaggregation, and Vitamin B-Producing Ability for New Probiotic Strain Introduction"

_microorganisms, 2022, doi:10.3390/microorganisms10020470_

Round 1

Reviewer 1 Report

The manuscript under review raises an interesting topic and describes a promising approach in modeling intestinal condition. However, there are some doubts about application of data obtained in vivo.

The authors claim that their data can be used in the production of fortified dairy products, as well as probiotics in vitamin deficiency. However, to study vitamin production, the authors use Folic Acid Assay Medium. Folic Acid Assay Medium is recommended for the microbiological assay of Folic Acid and is high in dextrose and free amino acids. Milk and intestinal contents have a completely different composition. It is known that the composition of the nutrient medium significantly affects the profile of substances secreted by the cell. How can the authors extrapolate data obtained on this medium to intestinal contents and milk?

There are also some minor remarks, such as:

line 61  but also have economically beneficial. - are economically beneficial?

Line 67 serum folate and homocysteine,  - levels of  serum folate and homocysteine?

Line 203 Bifidobacterium and Lactobacillus strains to simulated gastric juice - genus names should be in italics

Lines 224, 240 the same

Line 368 otherÑ‹ into fermentation broths  - others

Author Response

Dear collegue,

We are grateful for valuable comments.  Please see the ansvers in the attachment.

Reviewer 2 Report

The article deals with the characterisation of certain strains of Bifidobacterium and Lactobacillus regarding their probiotic potential using in vitro tests, and the possible production of vitamins of B group by the same strains. 

The study has a potential however, in its current form needs extensive revision. 

Some specific comments are listed below:

  1. In keywords please write the names of the genera in italics. Also correct activity to ability.
  2. Lines 39-42: The lack of all B vitamins can cause these conditions? You need to be specific. 
  3. Line 53 and 72: Correct "lactobacteria". Is not an appropriate word.
  4. Line 57: Generally Recognised As Safe. Please correct.
  5. Line 60: Its mg, not mcg.
  6. Line 61-62: Please rephrase.
  7. Line 79: Correct to aggregation ability. Same in line 85.
  8. Line 84: Remove "that".
  9. Line 100-103: Is it in g? Please include it. Same for line 108-114. 
  10. Also include in the materials and methods the names of all the manufacturing brands and the product codes. Same in line 105.
  11. Line 124-126: Not clear.
  12. Why up to 90 min in simulated gastric juice? Is it realistic?
  13. Why up to 3h in duodenum environment? Is it realistic?
  14. When was the bile medium used? This info is not provided in lines 115-126.
  15. Section 2.2.4.:Why tolerance was tested separately for these enzymes? Wouldn't it be more realistic to have included these enzymes in the previously tested juices?
  16. Line 169: Spectrophotometry.
  17. Line 185: Include the name of Software used.
  18. Line 191: What is Bc?
  19. Section 2.2.4.: How long was the exposure? This info is missing.
  20. Line 203: Italics missing.
  21. Line 205: Simulated.
  22. Line 212-213: After 40 min? You only checked 0, 30, 60, 90 min.
  23. Did you have any controls in the study?
  24. Figure 1: Its a wrong way to present survival rates. You can either put them in a table or translate the survival rates to actual counts with bars (initial-final)? Also, as it is, the x-axis needs more points.
  25. The caption in Figure 1 mentions a 0% at 60 min and 0% at 40 min. You did not check for 40 min. Where is this info shown in the figure?
  26. Line 224: Italics. 
  27. Figure 2 needs to be changed according to Figure 1. Also the pH was 6 or 7? Please correct.
  28. Statistical analysis was performed in the data? Statistical info is missing in all experiments. You cannot discuss your findings without it. 
  29. How many biological and technical replicates were performed for each test?
  30. Any comments related to the findings in gastric resistance for Lactobacillus and Bifidobacteria? Was these results expected?
  31. Figure 3 and 4 need to be changed similarly to 1 and 2. Captions also need to be more informative. including concentrations and exposure times. 
  32. In the conclusion section you need to specifically address any potential conclusions based on your findings. Conclusions cannot be that general. So was your initial hypothesis confirmed? Are these strains potentially probiotic? Can they be used for vitamin production? Which of these strains can be used and for the production of which vitamins?
  33. Also, in the materials and methods section you need to give some information about the origin of the tested strains? Any information which exists about them would help. For example, origin, product of isolation, with what criteria were selected for this study etc. 

Author Response

Dear coligue,

We are grateful for the painstaking work with our manuscript and tried to improve it according to the comments. Please see the answers in the attached file.

Round 2

Reviewer 2 Report

The authors tackled most of the issues identified in the previous report, however certain parts still require attention or are not clear at all:

-Correct mcg to mg throughout the manuscript.

-Why different sampling times were used for bifidobacteria and lactobacilli? The findings cannot be compared since the times are different. 

-Product coded should also be given for all materials used (not just the brand name).

-Triplicates you mean biological replicates? What about the technical replicates?

-In some cases the SD is very small. How can you explain this?

-Regarding controls, in similar studies certain reference probiotic bacteria have been used, such as Lb. rhamnosus GG or Lb. casei Shirota. See a couple of publications: Botta, C., Langerholc, T., Cencic, A, & Cocolin, L. (2014); Argyri, A.A., Zoumpopoulou, G., Karatzas, K.A.G., Tsakalidou, E., Nychas, G-J.E., Panagou, E.Z., & Tassou, C.C. (2013).

-The outcome of the statistical analysis is not found in the tables or the discussion. 

Author Response

Dear collegue,

We we have corrected the manuscript in accordance with your comments. Pleas see the responses in the attachment.
